# SARM1 depletion rescues NMNAT1-dependent photoreceptor cell death and retinal degeneration

Yo Sasaki[1]*, Hiroki Kakita[1,2], Shunsuke Kubota[3], Abdoulaye Sene[3], Tae Jun Lee[3], Norimitsu Ban[3], Zhenyu Dong[3], Joseph B Lin[3], Sanford L Boye[4], Aaron DiAntonio[5,6], Shannon E Boye[7], Rajendra S Apte[3,5,8]*, Jeffrey Milbrandt[1,6]

[1]Department of Genetics, Washington University School of Medicine, St. Louis, United States; [2]Department of Perinatal and Neonatal Medicine, Aichi Medical University, Aichi, Japan; [3]Department of Ophthalmology and Visual Sciences, Washington University School of Medicine, St. Louis, United States; [4]Department of Pediatrics, Powell Gene Therapy Center, Gainesville, United States; [5]Department of Developmental Biology, Washington University School of Medicine, St. Louis, United States; [6]Needleman Center for Neurometabolism and Axonal Therapeutics, St. Louis, United States; [7]Department of Pediatrics, Division of Cellular and Molecular Therapy, Gainesville, United States; [8]Department of Medicine, Washington University School of Medicine, St. Louis, United States

**Abstract** Leber congenital amaurosis type nine is an autosomal recessive retinopathy caused by mutations of the NAD+ synthesis enzyme NMNAT1. Despite the ubiquitous expression of NMNAT1, patients do not manifest pathologies other than retinal degeneration. Here we demonstrate that widespread NMNAT1 depletion in adult mice mirrors the human pathology, with selective loss of photoreceptors highlighting the exquisite vulnerability of these cells to NMNAT1 loss. Conditional deletion demonstrates that NMNAT1 is required within the photoreceptor. Mechanistically, loss of NMNAT1 activates the NADase SARM1, the central executioner of axon degeneration, to trigger photoreceptor death and vision loss. Hence, the essential function of NMNAT1 in photoreceptors is to inhibit SARM1, highlighting an unexpected shared mechanism between axonal degeneration and photoreceptor neurodegeneration. These results define a novel SARM1-dependent photoreceptor cell death pathway and identifies SARM1 as a therapeutic candidate for retinopathies.

*For correspondence:
sasaki@wustl.edu (YS);
apte@wustl.edu (RSA)

## Introduction

Leber congenital amaurosis (LCA) is a retinal degenerative disease characterized by childhood onset and severe loss of vision. LCA is the most common cause of blindness in children and about 70% of LCA cases are associated with mutations in genes related to the visual cycle, cGMP production, ciliogenesis, or transcription. Recently, more than thirty mutations in the nuclear NAD+ biosynthetic enzyme NMNAT1 were identified in patients with autosomal recessive LCA type 9 (LCA9) (*Falk et al., 2012*; *Perrault et al., 2012*; *Koenekoop et al., 2012*; *Chiang et al., 2012*; *Coppieters et al., 2015*; *Khan et al., 2018*). Despite the ubiquitous expression of this key NAD+ biosynthesis enzyme, LCA9 patients have no other systemic deficits outside the retina. In many cases, LCA9 associated mutant NMNAT1 proteins retain enzymatic activity and other biochemical functions, but appear to be less stable under conditions associated with cell stress (*Sasaki et al., 2015*). While it is clear that NAD+ deficiency in the retina is an early feature of retinal degenerative

disorders in mice (*Zabka et al., 2015*; *Lin et al., 2016*), it is not known which cell types and biological pathways are primarily affected in LCA9.

NMNAT1 plays important roles in diverse retinal functions. Overexpression of NMNAT1 in mouse retinal ganglion cells (RGCs) robustly protects against ischemic and glaucomatous loss of the axon and soma (*Zhu et al., 2013*), while conditional ablation in the developing mouse retina causes severe retinal dystrophy and loss of retinal function (*Eblimit et al., 2018*; *Wang et al., 2017*). Mice harboring *Nmnat1* mutations (V9M and D243G) exhibit severe retinal degeneration while the most common LCA9 mutation (E257K), which is not fully penetrant (*Siemiatkowska et al., 2014*), induces a milder retinal degeneration phenotype (*Eblimit et al., 2018*; *Greenwald et al., 2016*). In retinal explants, NMNAT1 promotes the survival of mouse retinal progenitor cells (*Kuribayashi et al., 2018*). The requirement for NMNAT in retina is evolutionarily conserved, as the *Drosophila* NMNAT isoform, dNMNAT, is required for the survival of photoreceptor cells after exposure to intense light (*Zhai et al., 2006*; *Zhai et al., 2008*).

The selective loss of photoreceptor cells in LCA9 suggests the survival and function of these cells are extremely sensitive to deranged $NAD^+$ metabolism. Indeed, many of the enzymes involved in photoreceptor function are dependent on $NAD^+$ as a cofactor, and for some of these proteins mutations in their corresponding genes lead to blindness. These include variants in the $NAD^+$ or NADPH dependent retinal dehydrogenases like RDH12 that cause LCA13 (*Schuster et al., 2007*) and the GTP synthesis enzyme IMPDH1 that causes retinitis pigmentosa (*Kennan et al., 2002*; *Bowne et al., 2002*). SIRT3, the mitochondrial $NAD^+$-dependent deacetylase is also important for photoreceptor homeostasis (*Lin et al., 2016*; *Lin et al., 2019*). Together, these observations highlight the importance of cytosolic $NAD^+$ dependent pathways in retinal function (*Lin et al., 2016*; *Lin and Apte, 2018*); however, the molecular roles of nuclear $NAD^+$ and NMNAT1 in the retina are largely unknown.

Multiple enzymatic pathways utilizing distinct metabolic precursors participate in $NAD^+$ biosynthesis (*Verdin, 2015*). However, in each case, these pathways converge at an NMNAT-dependent step that generates either $NAD^+$ or its deamidated form NaAD from the precursor NMN or NaMN. Among the three mammalian NMNAT isoforms, NMNAT1 is the only enzyme localized to the nucleus (*Berger et al., 2005*). However, in photoreceptors NMNAT1 is present in photoreceptor outer segments (*Zhao et al., 2016*), consistent with an additional, extra-nuclear role of NMNAT1 in photoreceptor cells. This is of particular interest because engineered non-nuclear variants of enzymatically-active NMNAT1 can potently inhibit pathological axon degeneration, which is commonly observed in the early stages of many neurodegenerative disorders (*Walker et al., 2017*; *Sasaki et al., 2009*; *Babetto et al., 2010*). When NMNAT1 is present in the axon, it can compensate for the injury-induced rapid loss of NMNAT2, the endogenous axonal NMNAT (*Gilley and Coleman, 2010*). NMNAT2 in turn, inhibits SARM1, an inducible $NAD^+$ cleavage enzyme (NADase) that is the central executioner of axon degeneration (*Gilley and Coleman, 2010*; *Gerdts et al., 2015*; *Sasaki et al., 2016*; *Gilley et al., 2015*; *Figley and DiAntonio, 2020*). Hence, mutations in NMNAT1 may promote retinal degeneration through the direct impact on $NAD^+$ biosynthesis and/or through the regulation of the SARM1-dependent degenerative program.

In this study, we determined the cell types and molecular mechanisms that cause retinal degeneration in LCA9. Using NMNAT1 conditional mutant mice, we showed that photoreceptors degenerate rapidly after the loss of NMNAT1 and that depletion of NMNAT1 in rod or cone cells is necessary and sufficient for the retinal degeneration. The AAV-mediated gene replacement of NMNAT1 in photoreceptors partially rescues the visual impairment caused by loss of NMNAT1. Finally, we determined the mechanism by which loss of NMNAT1 leads to photoreceptor degeneration. Loss of NMNAT1 leads to activation of SARM1 in photoreceptors, much as loss of NMNAT2 leads to SARM1 activation in axons (*Gilley and Coleman, 2010*). Moreover, photoreceptor degeneration is mediated by SARM1 in the absence of NMNAT1, much as axon degeneration and perinatal lethality is mediated by SARM1 in the absence of NMNAT2 (*Gilley and Coleman, 2010*; *Gilley et al., 2015*). Hence, photoreceptor neurodegeneration in LCA9 shares a deep mechanistic similarity to the pathological axon degeneration pathway. Since the SARM1 pathway is likely druggable (*DiAntonio, 2019*; *Krauss et al., 2020*), these findings provide a framework for developing new therapeutic strategies for treating patients with LCA9 and potentially other retinal disorders.

# Results

NMNAT1 is a nuclear enzyme that synthesizes NAD$^+$, an essential metabolite that is central to all aspects of cellular metabolism. NMNAT1 is indispensable for mouse development (*Conforti et al., 2011*) and recent studies identified causative mutations in *NMNAT1* in patients with Leber congenital amaurosis type 9 (LCA9), a disorder associated with severe, early-onset retinal degeneration and vision loss (*Falk et al., 2012*; *Perrault et al., 2012*; *Koenekoop et al., 2012*; *Chiang et al., 2012*; *Coppieters et al., 2015*; *Khan et al., 2018*). Patients with LCA9 have no systemic involvement outside the eye, suggesting that certain cells within the retina are particularly vulnerable to the loss of NMNAT1 function. Since no specific antibodies exist for immunocytochemical analysis of NMNAT1 localization, we determined its expression pattern in the retina using mice expressing an NMNAT1-lacZ fusion protein without the nuclear localization signal. Mice heterozygous for this mutant allele were viable and were used to map NMNAT1 expression by staining retinal sections with X-gal. LacZ staining was detected in the retinal pigment epithelium (RPE), photoreceptor outer segments (OS), inner segments (IS), outer nuclear layer (ONL), outer plexiform layer (OPL), inner nuclear layer (INL), inner plexiform layer (IPL), and ganglion cell layer (GCL) suggesting the ubiquitous expression of NMNAT1 in retina (*Figure 1—figure supplement 1A*).

LCA9 patients are mutant for NMNAT1 throughout the body, yet their defects are limited to the eye. In an effort to model this, we generated a global knockout using *Nmnat1$^{fl/fl}$: CAG-CreERT2* mice harboring homozygous *Nmnat1* floxed alleles (*Nmnat1$^{fl/fl}$*) and *CAG-CreERT2*, which expresses a tamoxifen-activated Cre recombinase from the ubiquitous actin promoter. We chose a conditional approach because NMNAT1 knockout embryos are lethal (*Conforti et al., 2011*). We treated 2-month-old *Nmnat1$^{fl/fl}$: CAG-CreERT2* and control mice with tamoxifen. We first used RT-PCR to measure *Nmnat1* mRNA in the retina at 21 days after tamoxifen and found that it was significantly decreased in NMNAT1 cKO (*Nmnat1$^{fl/fl}$: CAG-CreERT2* + tamoxifen) compared with wild-type (WT) mice (*Figure 1—figure supplement 1B*). To investigate the metabolic consequence of NMNAT1 deletion, we measured the levels of NMN, the substrate for NMNAT1, and NAD$^+$, the product of NMNAT1, in the retina at 25 days after tamoxifen injection. There is a significant increase in levels of NMN, presumably because it cannot be consumed by NMNAT1. There is also a mild decrease in NAD$^+$ in NMNAT1 cKO mice, although this is not statistically significant, suggesting that other NMNAT enzymes are an additional source of NAD$^+$ (*Figure 1A,B*). We next evaluated retinal pathology at 4 weeks after *Nmnat1* excision using biomicroscopy. Fundus images showed abnormalities including attenuation of blood vessels (*Figure 1C,D* arrowhead) and the appearance of a honeycomb structure, suggesting exposure of retinal pigment epithelium (RPE) cells (*Figure 1C,D* arrow) in the mutant animals. Histopathological examination of the retina with hematoxylin and eosin (HE) stained sections showed severe retinal degeneration as evidenced by the reduction of the retina thickness and the thinning of the outer nuclear layer (ONL) at 4 weeks post tamoxifen treatment (*Figure 1E,F*). Quantitative analysis demonstrated a significant reduction of retinal thickness, especially of the ONL (*Figure 1G,H*). Hence, photoreceptor cells are highly vulnerable following the loss of NMNAT1.

To gain insights into the temporal aspects of the retinal degenerative process, we analyzed retinal morphology at seven time points after tamoxifen administration. The loss of nuclei in the ONL layers were evident at 25 days post tamoxifen injection and robust retinal thinning was evident at 33 days post tamoxifen injection (*Figure 2A*). We measured the loss of photoreceptor cells by counting the number of ONL cell nuclei. Cell loss was first detected in the ONL around 3 weeks after tamoxifen administration and gradually progressed such that only ~15% of the cells remained at 33 days (*Figure 2B*).

Next, we evaluated retinal function after NMNAT1 deletion using electroretinogram (ERG). We examined three cohorts of mice: *Nmnat1$^{fl/fl}$: CAG-CreERT2* treated with tamoxifen, untreated *Nmnat1$^{fl/fl}$: CAG-CreERT2* or *Nmnat1$^{fl/fl}$* treated with tamoxifen. In mutant animals in which *Nmnat1* was excised, we observed a complete loss of both scotopic (rod-driven responses) and photopic (cone-driven responses) responses, indicating the loss of *Nmnat1* in mature retina causes severe photoreceptor dysfunction (*Figure 2C–E*). This is consistent with previous reports showing developmental retinal defects in the tissue specific *Nmnat1* knockout mice (*Eblimit et al., 2018*; *Wang et al., 2017*). While previous reports show that *Nmnat1* is necessary for appropriate retinal development, our pathological and functional analyses of conditional deletion of NMNAT1 in two-

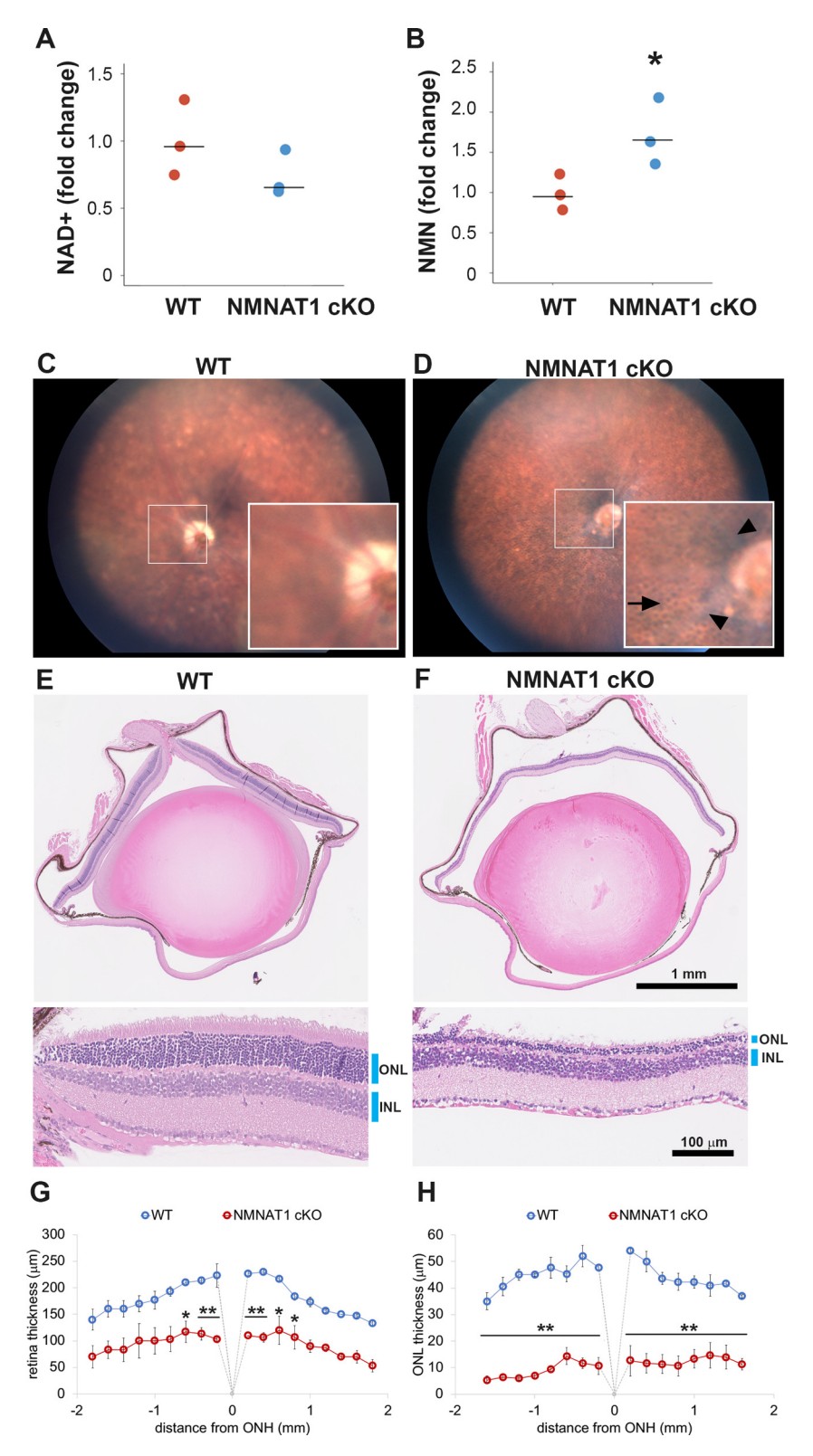

**Figure 1.** NMNAT1 depletion induces severe retinal degeneration. (**A, B**) Metabolite analysis by LC-MSMS in retinal tissues from WT or NMNAT1 conditional knockout (*Nmnat1^{fl/fl}: CAG-CreERT2* + tamoxifen: NMNAT1 cKO) mice at 25 days post tamoxifen injection. Fold changes of NAD$^+$ (**A**) and NMN (**B**) concentrations compared with that of WT retinal tissues are shown. *p<0.05 denotes the significant difference from WT with Kruskal-Wallis test (n = 3 mice for WT and n = 3 mice for NMNAT1 cKO). Graphs show the all data points and median (cross bars). (**C, D**) Fundus biomicroscopy

*Figure 1 continued on next page*

*Figure 1 continued*

images of the retina from wild type (WT, **C**) or NMNAT1 conditional knock out (*Nmnat1$^{fl/fl}$: CAG-CreERT2* + tamoxifen: NMNAT1 cKO, **D**) mice at 4 weeks post tamoxifen injection. (**E, F**) representative images of hematoxylin and eosin stained eye sections from WT mice (**E**) or NMNAT1 conditional knockout (*Nmnat1$^{fl/fl}$: CAG-CreERT2* + tamoxifen: NMNAT1 cKO, **E**) mice at 4 weeks post tamoxifen injection (ONL: outer nuclear layer and INL: inner nuclear layer). The substantial thinning of the ONL was observed in 3 WT and 3 NMNAT1 cKO mice. (**G**) The quantification of the retina thickness from WT and NMNAT1 conditional knockout (*Nmnat1$^{fl/fl}$: CAG-CreERT2* + tamoxifen: NMNAT1 cKO) mice were shown. Graphs show the average and error bars represent the standard error. Statistical analysis was performed by two-way ANOVA with Tukey post-hoc test (n = 3 mice for WT, n = 3 mice for NMNAT1 cKO (*Nmnat1$^{fl/fl}$: CAG-CreERT2* + tamoxifen at 4 weeks post tamoxifen injection)). $F_{(1, 72)}=309$, $p<1.0\times10^{-16}$ between WT and NMNAT1 cKO retina. *$p<0.05$ and **$p<0.001$ denotes the significant difference compared with WT retina. (**H**) The quantification of the ONL thickness from WT and NMNAT1 conditional knockout (*Nmnat1$^{fl/fl}$: CAG-CreERT2* + tamoxifen: NMNAT1 cKO) mice were shown. Graphs show the average and error bars represent the standard error. Statistical analysis was performed by two-way ANOVA with Tukey post-hoc test (n = 3 mice for WT and n = 3 mice for NMNAT1 cKO). $F_{(1, 72)}=1023$, $p<1.0\times10^{-16}$ between WT and NMNAT1 cKO retina. **$p<0.001$ denotes the significant difference compared WT.

The online version of this article includes the following source data and figure supplement(s) for figure 1:

**Source data 1.** Source data for *Figure 1*.
**Figure supplement 1.** NMNAT1 is ubiquitously expressed in the retina.
**Figure supplement 1—source data 1.** Source data for *Figure 1—figure supplement 1*.

month-old mice demonstrates that NMNAT1 is also necessary for photoreceptor cell maintenance and mature retinal functions.

In addition to NMNAT1, mammalian cells encode two other NMNAT isoforms; NMNAT2 that is localized in the Golgi and cytosol, and NMNAT3 that is localized inside the mitochondria. Since the loss of NMNAT1 induced retinal degeneration, we wished to determine the role of NMNAT2 and 3 in the retinal structure/function. A previous study showed that NMNAT2 knockout mice are perinatally lethal and have truncated optic nerves as well as peripheral axon degeneration (*Slivicki et al., 2016*). We could not assess the role of NMNAT2 in retinal function due to the lack of conditional knockout mice. On the other hand, NMNAT3 deficient mice (NMNAT3 KO) are viable with splenomegaly and hemolytic anemia (*Hikosaka et al., 2014*). We generated NMNAT3 KO mice and investigated their retinal function using ERG. Consistent with the previous report, NMNAT3 KO mice showed splenomegaly (data not shown), however, there were no defects in ERG (*Figure 2—figure supplement 1*). These results indicate that NMNAT3 is dispensable for retinal function, suggesting NMNAT1 is the functionally dominant isoform controlling retinal phenotype.

Identifying the cells that are vulnerable to NMNAT1 loss is key to understanding LCA9 pathogenesis. The severe loss of the ONL nuclei (*Figure 2B*) induced by NMNAT1 deletion prompted us to test whether loss of NMNAT1 specifically in photoreceptors would result in their death and recapitulate the phenotype observed using the widely expressed *CAG-CreERT2*. We therefore generated mice lacking NMNAT1 specifically in rod photoreceptors by crossing the *Nmnat1$^{fl/fl}$* mice with *Rhodopsin-Cre* (*Rho-Cre*) mice (*Li et al., 2005*). We analyzed the retinas of *Nmnat1$^{fl/fl}$:Rho-Cre* mice at 6-weeks-of-age. Similar to previous findings using *Crx-Cre* that expresses Cre recombinase in developing photoreceptors as early as E11 (*Eblimit et al., 2018*; *Wang et al., 2017*), histological analysis revealed severe thinning of the ONL in these mutant mice (*Figure 3A,B*). The quantitative analysis showed a significant reduction of the retina and ONL thickness in *Nmnat1$^{fl/fl}$:Rho-Cre* retina (*Figure 3D*) as well as a significant reduction in ONL cell number as detected by nuclear counts (*Figure 3F*). Consistent with the loss of ONL cells, ERG analysis showed a severe reduction in the scotopic a- and –b-waves, representing rod photoreceptor function, in the *Nmnat1$^{fl/fl}$:Rho-Cre* mice (*Figure 3G,H*). In addition, we found decreases in cone mediated photoresponses (photopic b-wave signal) (*Figure 3I*) that is likely secondary to a loss of rod photoreceptor cells due to loss of required rod-derived survival factors (*Lin et al., 2016*; *Aït-Ali et al., 2015*).

To explore directly the role of NMNAT1 in cones, we deleted NMNAT1 using the cone-specific *OPN1LW-Cre*. We crossed *Nmnat1$^{fl/fl}$* mice with *OPN1LW-Cre* mice in which Cre recombinase expression is driven by the human red/green pigment (*OPN1LW*) promoter starting at P10 (*Le et al., 2004*). At 6-weeks-of-age we examined these mutant mice histologically, but did not detect any gross abnormalities, presumably due to the low number of cones (only 3% of total photoreceptors) in mice (*Figure 3A,C,E*). However, ERG analysis showed a complete loss of the photopic b-wave, which is derived from cone photoreceptors. This functional result demonstrates that NMNAT1 activity is vital for cone function (*Figure 3I*). In summary, these genetic ablation experiments demonstrate

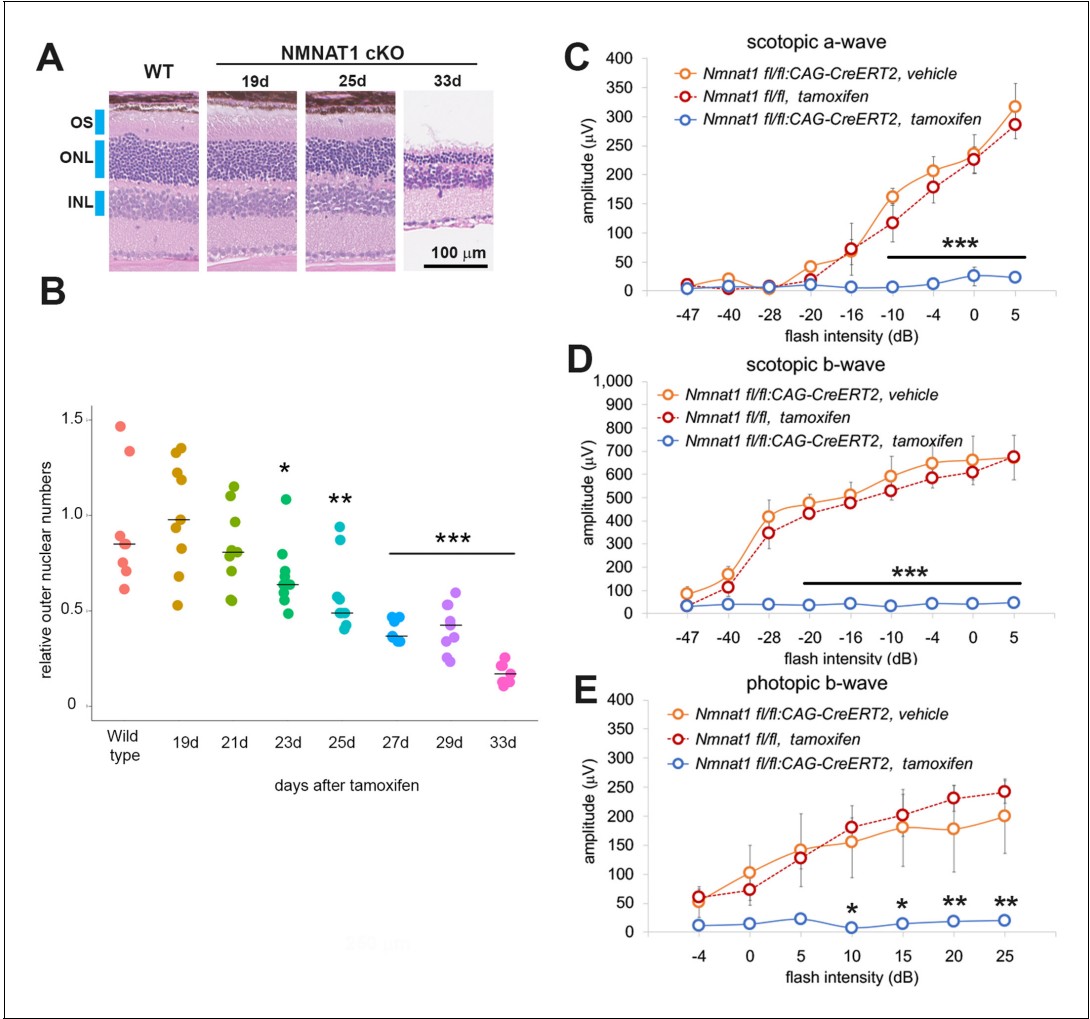

**Figure 2.** NMNAT1 induces the loss of photoreceptor cells and retinal function. (**A**) Representative images of hematoxylin and eosin stained sections showing time course of retinal degeneration in NMNAT1 conditional knockout (*Nmnat1*$^{fl/fl}$*: CAG-CreERT2* + tamoxifen: NMNAT1 cKO) mice at 19 to 33 days post tamoxifen injection or littermate wild-type (WT) mice at 33 days post tamoxifen injection (WT). Blue bars indicate outer nuclear layer (ONL), inner nuclear layer (INL), and outer segment (OS). Similar results were obtained from three mice at each time point. (**B**) Quantification of relative ONL nuclei numbers of NMNAT1 conditional knockout mouse (*Nmnat1*$^{fl/fl}$*: CAG-CreERT2* + tamoxifen: NMNAT1 cKO) compared with WT at various time after tamoxifen injection. The graph shows all data points and median (cross bars). Statistical analysis was performed by one-way ANOVA with Holm-Bonferroni multiple comparison (n = 3 mice for each of WT, 19d, 21d, 33d and n = 4 mice for each of 25d, 27d). $F_{(7, 64)}=19$, $p=1.9 \times 10^{-13}$. *p<0.05, **p<0.001, and ***p<0.0001 denotes the significant difference compared with WT. (**C, D, E**) ERG analysis of controls (*Nmnat1*$^{fl/fl}$*: CAG-CreERT2* vehicle or *Nmnat1*$^{fl/fl}$ + tamoxifen) and NMNAT1 conditional knockout (*Nmnat1*$^{fl/fl}$*: CAG-CreERT2* + tamoxifen: NMNAT1 cKO). Graphs show the average and error bars represent the standard error. Scotopic a-wave (**C**), scotopic b-wave (**D**), and photopic b-wave (**E**) are shown. Statistical analysis was performed by two-way ANOVA with Tukey post-hoc test (n = 3 mice for *Nmnat1*$^{fl/fl}$*: CAG-CreERT2* with vehicle, n = 3 mice for *Nmnat1*$^{fl/fl}$ at 33 days post tamoxifen injection, n = 4 mice for *Nmnat1*$^{fl/fl}$*: CAG-CreERT2* at 33 days post tamoxifen injection). $F_{(1, 72)}=220$, $p<2 \times 10^{-16}$ between controls (*Nmnat1*$^{fl/fl}$*: CAG-CreERT2* with vehicle and *Nmnat1*$^{fl/fl}$ 33 days post tamoxifen injection) and NMNAT1 cKO for scotopic a-wave, $F_{(1, 72)}=633$, $p<2 \times 10^{-16}$ between controls and NMNAT1 cKO for scotopic b-wave, $F_{(1, 56)}=94$, $p=1.3 \times 10^{-13}$ between controls and NMNAT1 cKO for photopic b-wave. *p<0.05, **p<0.001, and ***p<0.0001 denote a significant difference compared with WT.

The online version of this article includes the following source data and figure supplement(s) for figure 2:

**Source data 1.** Source data for *Figure 2*.
**Figure supplement 1.** ERG analysis of NMNAT3-deficient retina.
**Figure supplement 1—source data 1.** Source data for *Figure 2—figure supplement 1*.

the importance of NMNAT1 for proper function and survival of both rods and cones, and indicate that LCA9-associated retinal degeneration is likely due to the direct cell-autonomous effects of *NMNAT1* mutations in photoreceptors.

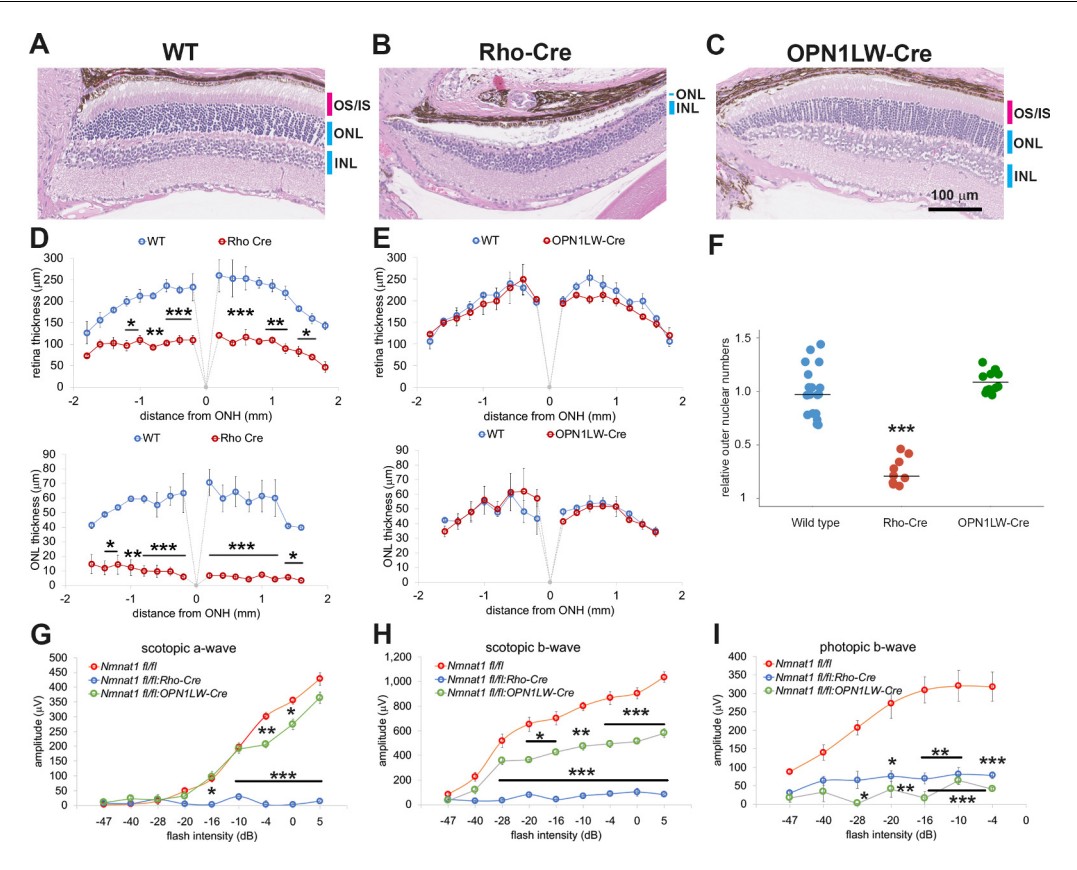

**Figure 3.** Photoreceptor-specific depletion of NMNAT1 induces retinal degeneration. (A, B, C) Hematoxylin and eosin stained eye sections from 6 week old wild-type (WT, **A**), rod-specific NMNAT1 KO (*Nmnat1^{fl/fl}:Rho-Cre*: Rho-Cre, **B**), or cone-specific NMNAT1 KO (*Nmnat1^{fl/fl}:OPN1LW-Cre*: OPN1LW-Cre, **C**) mice. Blue bars indicate outer nuclear layer (ONL) and inner nuclear layer (INL). Red bars indicate the outer segments (OS) and inner segments (IS). Similar results were obtained from three mice for each genotype. (**D**) Quantification of retina and ONL thickness in WT or rod-specific NMNAT1 KO (Rho-Cre) retinas. The graph shows all data points and median (cross bars). Statistical analysis was performed by two-way ANOVA with Tukey post-hoc test (n = 3 mice for WT and n = 3 mice for Rho-Cre). $F_{(1, 72)}=428$, $p<2\times10^{-16}$ between WT and Rho-Cre retina thickness and $F_{(1, 64)}=530$, $p<2\times10^{-16}$ between WT and Rho-Cre ONL thickness. \*$p<0.05$, \*\*$p<0.001$, and \*\*\*$p<0.0001$ denote significant differences compared with WT. (**E**) Quantification of retina and ONL thickness in WT or cone-specific NMNAT1 KO (OPN1LW-Cre) retinas. Graphs show the average and error bars represent the standard error. Statistical analysis was performed by two-way ANOVA with Tukey post-hoc test (n = 3 mice for WT and n = 3 mice for OPN1LW-Cre). $F_{(1, 72)}=4$, $p=0.037$ between WT and OPN1LW-Cre retina thickness and $F_{(1, 64)}=0.03$, $p=0.87$ between WT and OPN1LW-Cre ONL thickness. There are no significant differences in OPN1LW-Cre retina and ONL thickness compared with WT. (**F**) Quantification of relative ONL nuclei numbers compared with WT. The graph shows all data points and median (cross bars). Statistical analysis was performed by one-way ANOVA with Holm-Bonferroni multiple comparison (n = 6 mice for WT, n = 3 mice for Rho-Cre, n = 3 mice for OPN1LW-Cre). $F_{(2, 35)}=59$, $p=5.9\times10^{-12}$. \*\*\*$p<0.0001$ denotes the significant difference compared WT. (**G, H, I**) ERG analysis of WT, Rho-Cre, and OPN1LW-Cre mice. Scotopic a-wave (**G**), scotopic b-wave (**H**), and photopic b-wave (**I**) are shown. Graphs show the average and error bars represent the standard error. Statistical analysis was performed by two-way ANOVA with Tukey post-hoc test (n = 6 mice for WT, n = 3 mice for Rho-Cre, n = 3 mice for OPN1LW-Cre). $F_{(2, 81)}=314$, $p<2.0\times10^{-16}$ among genotypes (WT, Rho-Cre, and OPN1LW-Cre) for scotopic a-wave, $F_{(2, 81)}=413$, $p<2\times10^{-16}$ among genotypes for scotopic b-wave, $F_{(2, 63)}=102$, $p<2\times10^{-16}$ among genotypes for photopic b-wave. \*$p<0.05$, \*\*$p<0.001$, and \*\*\*$p<0.0001$ denote a significant difference compared with WT.

The online version of this article includes the following source data and figure supplement(s) for figure 3:

**Source data 1.** Source data for *Figure 3*.
**Figure supplement 1.** AAV-NMNAT1 partially rescued the retinal degeneration in NMNAT1-deficient retinas.
**Figure supplement 1—source data 1.** Source data for *Figure 3—figure supplement 1*.

The loss-of-function studies above demonstrate that NMNAT1 is necessary in photoreceptors for proper retinal function. Next we assessed whether viral-mediated expression of NMNAT1 in photoreceptors in an otherwise NMNAT1 deficient animal is sufficient to promote retinal function. First, we developed a system for retinal expression of transgenes. We subretinally delivered AAV8(Y733F) containing the photoreceptor-specific human rhodopsin kinase (hGRK1) promoter driving GFP

(*Kay et al., 2013*; *Boye et al., 2013*). Virus was injected into the subretinal space of two-month-old wild type mice. Transgene expression was evaluated 4–6 weeks post-injection. AAV-mediated GFP expression was observed in a subset of rhodopsin-positive cells but was weak in the inner nuclear layer (INL) (*Figure 3—figure supplement 1A*). These results confirm earlier reports that the hGRK1 promoter restricts transgene expression primarily to photoreceptors (*Kay et al., 2013*). We next asked whether AAV-mediated expression of HA-tagged human NMNAT1 could prevent retinal degeneration caused by *Nmnat1* excision. Two-month-old *Nmnat1^{fl/fl}; CAG-CreERT2 mice* received subretinal injections of AAV-NMNAT1 in one eye, and control vector (AAV-GFP) in the contralateral eye. We confirmed the expression of NMNAT1-HA in a subset of outer nuclear cells and a minor population of inner nuclear cells (*Figure 3—figure supplement 1B,C*). Mice that received AAV-NMNAT1 and AAV-GFP were then treated with tamoxifen to deplete endogenous NMNAT1. One month after tamoxifen treatment, we examined retinal function. Despite the expression of NMNAT1 in only a subset of photoreceptors, we observed significantly increased scotopic a-wave amplitudes in AAV-NMNAT1 treated retinas compared with retinas injected with AAV-GFP (*Figure 3—figure supplement 1D*; *Figure 3D*). There were also small, but statistically insignificant, increases in the scotopic and photopic b-wave amplitudes between AAV-NMNAT1 and AAV-GFP treated retinas (*Figure 3—figure supplement 1E,F*). Hence, NMNAT1 gene delivery to photoreceptor cells significantly improved their function in this LCA9 model.

We next sought to determine the molecular mechanisms required for retinal degeneration in the NMNAT1-deficient retina. In injured peripheral nerves, the loss of NMNAT2 induces an increase in NMN that is hypothesized to activate SARM1-dependent axon degeneration (*Di Stefano et al., 2015*; *Zhao et al., 2019*). Our metabolomic analysis revealed that NMN is increased in the NMNAT1-deficient retinas (*Figure 1B*), and previous studies have detected SARM1 in mouse and bovine photoreceptor cells (*Zhao et al., 2016*; *Datta et al., 2015*; *Menon et al., 2019*). These results raised the possibility that the increased retinal NMN activates SARM1 NADase activity, inducing NAD$^+$ loss and cellular degeneration in the retina. To test this hypothesis, we crossed *Nmnat1^{fl/fl}:CAG-CreERT2* mice with SARM1 knockout mice (*Szretter et al., 2009*) to generate *Nmnat1^{fl/fl}: CAG-CreERT2:Sarm1^{-/-}* mice. *Nmnat1* was excised in these mice via tamoxifen administration at 2 months of age (NMNAT1 cKO: SARM1 KO). First, we assessed SARM1 activation via measurement of cADPR, a product of the SARM1 NAD$^+$ cleavage enzyme and a biomarker of SARM1 activity as well as NAD$^+$ (*Sasaki et al., 2020*). While the loss of NAD$^+$ was not statistically significant at 25 days post tamoxifen injection in NMNAT1 cKO retina (*Figure 1A*), there was significant loss of NAD$^+$ at 29 to 32 days post tamoxifen in NMNAT1 cKO but not in NMNAT1 cKO: SARM1 KO retina (*Figure 4A*). These data suggest activation of the SARM1 NADase in NMNAT1-deficient retina. Consistent with this idea, we also observed a significant increase of cADPR in NMNAT1 cKO retina in a SARM1-dependent manner (*Figure 4B*). Hence, SARM1 is activated by the loss of NMNAT1. Next, we assessed retinal degeneration. While there is a dramatic loss of ONL nuclei 32 days post tamoxifen injection in NMNAT1 cKO retina, there was no obvious loss of ONL cells in NMNAT1 cKO: SARM1 KO retina (*Figure 4C–E*). Quantitative analysis showed no reduction of retinal and ONL thickness in NMNAT1 cKO:SARM1 KO retina compared with WT (*Figure 4F,G*) in sharp contrast to the dramatic loss of retinal and ONL thickness in the NMNAT1 cKO (*Figures 1* and *2*). Moreover, there was no detectable loss of ONL nuclei in NMNAT1 cKO:SARM1 KO compared with WT, demonstrating that SARM1 is necessary for photoreceptor cell death induced by the loss of NMNAT1 (*Figure 4H*). We next examined the functional consequences of NMNAT1 depletion in the presence or absence of SARM1 using ERGs, and again found that loss of SARM1 prevented the severe loss of both scotopic and photopic responses due to NMNAT1 deficiency (*Figure 4I–K*). Taken together, these findings demonstrate that loss of NMNAT1 leads to the activation of SARM1, and that SARM1 is required for the subsequent photoreceptor degeneration and loss of visual function. Therefore, the essential function of NMNAT1 in photoreceptors is to inhibit SARM1, and inhibition of SARM1 is a candidate therapeutic strategy for the treatment of LCA9.

## Discussion

In this study, we demonstrate that deletion of NMNAT1 in the adult retina causes a dramatic loss of photoreceptors and a concomitant reduction in retinal function. In addition, cell-type specific deletion of NMNAT1 in early postnatal photoreceptors is sufficient to induce retinal degeneration.

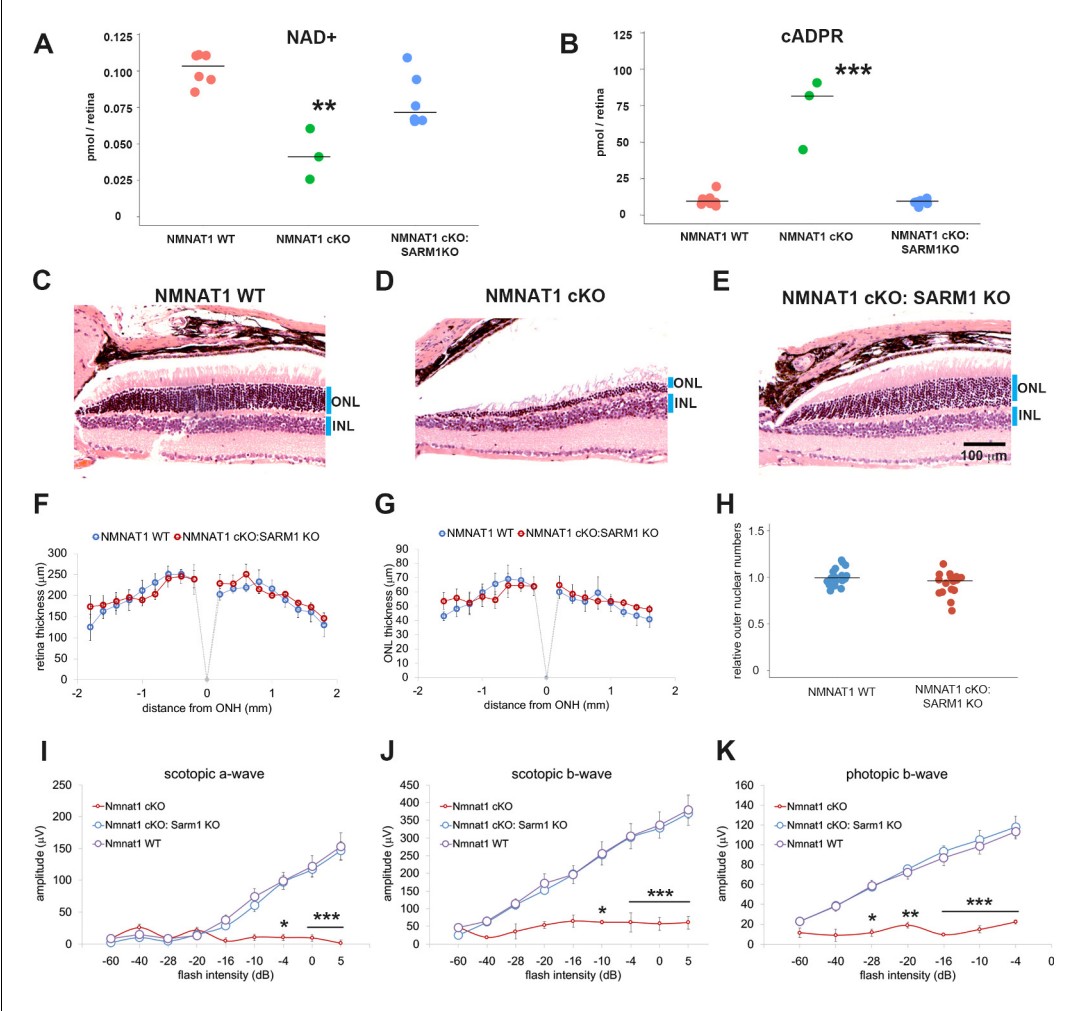

**Figure 4.** Depletion of SARM1 rescues retinal degeneration in the NMNAT1-deficient retina. (A, B) Metabolite analysis by LC-MSMS in retinal tissues from WT, NMNAT1 conditional knockout (*Nmnat1^{fl/fl}: CAG-CreERT2* + tamoxifen at 29 to 32 days post tamoxifen injection: NMNAT1 cKO), or NMNAT1 cKO:SARM1 KO mice were shown. Metabolites from whole retina of one eye were analyzed for $NAD^+$ (A) and cADPR (B) concentrations and compared with that of WT retinal tissues . Graphs show the all data points and median (cross bars). Statistical analysis was performed by one-way ANOVA with Holm-Bonferroni multiple comparison (n = 7 mice for WT, n = 3 mice for NMNAT1 cKO, and n = 6 mice for NMNAT1 cKO:SARM1 KO). $F(2, 13)=259$, $p=3.0\times10^{-4}$ for NAD+ and $F(2,13) = 48$, $p=9.43\times10^{-7}$ for cADPR. **p<0.001 and ***p<0.0001 denote the significant difference compared with WT. (C, D, E) Representative images of hematoxylin and eosin stained eye sections from NMNAT1 WT (*Nmnat1^{fl/fl}: Sarm1^{+/-}*, C), NMNAT1 cKO (*Nmnat1^{fl/fl}:Sarm1^{+/+}: CAG-CreERT2* post 32 days tamoxifen injection, D), and NMNAT1 cKO: SARM1 KO (*Nmnat1^{fl/fl}: Sarm1^{-/-}: CAG-CreERT2* at post 32 days tamoxifen injection, E). Blue bars represent outer nuclear layer (ONL) and inner nuclear layer (INL). Similar results were obtained from three mice for WT, two mice for NMNAT1 cKO, and three mice for NMNAT1cKO:SARM1 KO. (F) The quantification of the retina thickness from NMNAT1 WT and NMNAT1 cKO:SARM1 KO mice were shown. Graphs show the average and error bars represent the standard error. Statistical analysis was performed by two-way ANOVA with Tukey post-hoc test (n = 3 mice for NMNAT1 WT, n = 3 mice for NMNAT1 cKO:SARM1 dKO). $F(1, 72)=0.8$, $p=0.37$ between NMNAT1 WT and NMNAT1 cKO:SARM1 KO retina. There is no significant difference between NMNAT1 WT and NMNAT1 cKO:SARM1 KO. (G) The quantification of the outer nuclear layer (ONL) thickness from NMNAT1 WT and NMNAT1 cKO:SARM1 KO mice were shown. Graphs show the average and error bars represent the standard error. Statistical analysis was performed by two-way ANOVA with Tukey post-hoc test (n = 3 mice for NMNAT1 WT, n = 3 mice for NMNAT1 cKO:SARM1 KO). $F(1, 64)=0.43$, $p=0.51$ between NMNAT1 WT and NMNAT1 cKO:SARM1 KO retina. There is no significant difference between NMNAT1 WT and NMNAT1 cKO:SARM1 KO. (H) Quantification of relative ONL nuclei numbers compared with WT. The graph shows all data points and median (cross bars). Statistical analysis was performed by Mann-Whitney U test (n = 3 mice for NMNAT1 WT, n = 3 mice for NMNAT1 cKO:SARM1 KO). p=0.10. There are no statistical differences between NMNAT1 WT and NMNAT1 cKO:SARM1KO. (I, J, K) ERG analysis of NMNAT1 WT (*Nmnat1^{fl/fl}: Sarm1^{+/-}* or *Nmnat1^{fl/fl}: Sarm1^{-/-}*), NMNAT1 cKO (*Nmnat1^{fl/fl}: CAG-CreERT2* post 29 to 32 days tamoxifen injection), and NMNAT1 cKO: SARM1 KO (*Nmnat1^{fl/fl}: SARM1^{-/-}: CAG-CreERT2* post 32 days tamoxifen injection). Graphs show the average and error bars represent the standard error. Statistical analysis was performed by two-way ANOVA with Tukey post-hoc test (n = 8 mice for NMNAT1 WT, n = 3 mice for NMNAT1 cKO, and n = 8 mice for NMNAT1 cKO: SARM1 KO). $F(2, 144)=29$, $p=2.9\times10^{-11}$ among genotypes (NMNAT1 WT, NMNAT1 cKO, NMNAT1 cKO:SARM1 KO) for scotopic a-wave, $F(2, 144)=46$, $p<2.0\times10^{-16}$ among genotypes for scotopic b-wave, $F(2, 112)=94$, $p<2.0\times10^{-16}$ among

*Figure 4 continued on next page*

*Figure 4 continued*

genotypes for photopic b-wave. *p<0.05, **p<0.001, and ***p<0.0001 denotes the statistical difference between NMNAT1 WT and NMNAT1 cKO or between NMNAT1 cKO:SARM1KO and NMNAT1 cKO. There is no statistical difference between NMNAT1 WT and NMNAT1 cKO: SARM1 KO.

The online version of this article includes the following source data for figure 4:

**Source data 1.** Source data for *Figure 4*.

Hence, NMNAT1 is required for the survival and function of both developing and mature photoreceptors. Using a modified AAV8 vector and the human rhodopsin kinase promoter to express NMNAT1, we demonstrated that a gene replacement strategy can improve retinal function in this model of LCA9. Finally, we defined the molecular mechanism by which NMNAT1 promotes photoreceptor function and survival. In photoreceptors, loss of NMNAT1 leads to activation of the inducible NADase SARM1 and the SARM1-dependent degeneration of photoreceptors. This finding defines a common mechanism operant in both photoreceptor degeneration and pathological axon degeneration. Loss of NMNAT1 in photoreceptors or NMNAT2 in axons leads to the SARM1-induced death of photoreceptors or axons, respectively. This surprising result extends our understanding of both the mechanisms causing retinal degeneration and the potential role of SARM1 in human disease (*Figley and DiAntonio, 2020*; *Coleman and Höke, 2020*).

Retinal $NAD^+$ homeostasis is crucial for visual function and $NAD^+$ decline is a hallmark of many retinal degenerative disease models (*Lin et al., 2016*). Reduced $NAD^+$ induces mitochondrial dysfunction in photoreceptor cells and affects activity of SIRT3, which protects the retina from light-induced and other forms of neurodegeneration. In addition, $NAD^+$-dependent enzymes play crucial roles in phototransduction including the regeneration of the photosensitive element, 11-cis-retinal, and the regulation of photoreceptor membrane potential. Moreover, mutations in the genes encoding some of these enzymes cause retinal degenerative disease. For example, mutations in all-trans-retinal dehydrogenase (RDH12) that is localized to photoreceptor cells are associated with LCA13. Combined deletion of retinal dehydrogenases, RDH12 and RDH8, results in mouse retinal degeneration (*Maeda et al., 2009*). $NAD^+$ is also a cofactor for inosine monophosphate dehydrogenase (IMPDH1), which is the rate limiting enzyme for GTP synthesis and, in turn, is required for cGMP production. cGMP is indispensable for the regulation of photoreceptor membrane potential and calcium concentration upon light stimulation. IMPDH1 mutations cause both a dominant form of retinitis pigmentosa (RP10) and LCA11. These results highlight the central role of $NAD^+$ metabolism in the photoreceptor.

NMNAT1 is the only NMNAT enzyme localized to the nucleus in mammals and is crucial for nuclear $NAD^+$ synthesis. Despite the broad functions of nuclear $NAD^+$ in all cell types, the sole consequence of LCA9-associated *NMNAT1* mutations is retinal dysfunction/degeneration without systemic abnormalities. Previous studies, and our results, show early loss of photoreceptor cells in NMNAT1-deficient retina (*Eblimit et al., 2018*; *Wang et al., 2017*). In photoreceptors NMNAT1 may localize not only in the nucleus but also outside the nucleus, since a subcellular proteomics study showed the existence and enrichment of NMNAT1 in the photoreceptor outer segments (*Zhao et al., 2016*). Single-cell transcriptomic RNA analysis also found *Nmnat1* in rods and cones (*Menon et al., 2019*; *Lukowski et al., 2019*). Consistent with an extranuclear role for NMNAT1 in photoreceptors, in these studies cytosolic NMNAT2 was either not identified or was found at much lower levels than NMNAT1. NMNAT3 is the mitochondrial NMNAT, and we show here that it is dispensable for retinal homeostasis and function, further highlighting the central requirement for NMNAT1 in photoreceptors.

Having demonstrated that NMNAT1 is required in photoreceptors in this model of LCA9, we showed that viral-mediated gene replacement in photoreceptors is capable of improving retinal function. Adeno-associated virus (AAV) is a naturally occurring, non-pathogenic virus used in gene therapy studies to restore structure and function to diseased cells. Recently, the U.S. FDA approved an AAV-RPE65 vector as a therapeutic reagent for LCA2 and other biallelic RPE65 mutation associated retinal dystrophies (*Apte, 2018*). Theoretically, LCA9 caused by the loss of NMNAT1 function is a reasonable target for AAV-mediated gene therapy. To achieve expression in photoreceptors, we used an AAV8 variant that is a highly efficient for transducing photoreceptors following subretinal injection as well as the hGRK1 promoter that has activity exclusively in rods and cones (*Kay et al.,*

*2013*; *Boye et al., 2013*). Delivery of AAV8(Y733F)-hGRK1-NMNAT1 prior to deletion of NMNAT1 resulted in partial improvement of the retinal phenotype, in particular the scotopic a-wave, of mice deficient for NMNAT1. Future studies will assess the efficacy of gene replacement after deletion of NMNAT1 to more closely mimic the human condition.

Since $NAD^+$ plays such a central role in photoreceptors, the identification of the $NAD^+$ biosynthetic enzyme NMNAT1 as the cause of LCA9 suggests that photoreceptor degeneration in LCA9 is due the reduction in $NAD^+$ synthesis. Surprisingly, we demonstrate here that this is not the essential function for NMNAT1 in photoreceptors. Instead, NMNAT1 is required to restrain the activity of the prodegenerative NADase SARM1. When NMNAT1 is deleted from SARM1 KO photoreceptors, the photoreceptors do not die and but instead maintain their physiological function, demonstrating that these cells do not require NMNAT1 as long as SARM1 is not present. This finding is perfectly analogous to the relationship between NMNAT2 and SARM1 in the axon. NMNAT2 KO mice are perinatal lethal and have dramatic axonal defects, but NMNAT2, SARM1 double KO mice are viable and have a normal lifespan (*Gilley et al., 2015*). NMNAT enzymes inhibit the activation of SARM1 (*Sasaki et al., 2016*), potentially by consuming the $NAD^+$ precursor NMN, which is postulated to activate SARM1 (*Di Stefano et al., 2015*; *Zhao et al., 2019*). Prior to our current study, loss of NMNAT2 was the only known trigger of SARM1 activation. Our current work suggests that SARM1 is activated by the loss of any NMNAT enzyme whose activity is not redundant with another NMNAT isoform. NMNAT2 is the only cytosolic NMNAT in the axon, and so loss of axonal NMNAT2 leads to localized activation of SARM1 and axon degeneration. In photoreceptors, NMNAT1 is not only nuclear but also likely extranuclear, and NMNAT2 is apparently present at very low levels. Hence, in photoreceptors loss of NMNAT1 triggers activation of SARM1 which consumes $NAD^+$ and triggers cell death. As $NAD^+$ loss is a common pathology of many retinal diseases, this raises the possibility that SARM1 activation may contribute to a wide range of retinal disorders. In support of this conjecture, recent studies found that SARM1 promotes retinal degeneration in X-linked retinoschisis (*Molday et al., 2007*) and rhodopsin-deficient mice (*Ozaki et al., 2020*).

Our identification of SARM1 as the executioner of photoreceptor death in this model of LCA9 opens up new therapeutic possibilities. We previously developed a potent dominant negative SARM1 variant and demonstrated that AAV-mediated expression of dominant negative SARM1 strongly protects injured axons from degeneration in the peripheral nervous system (*Geisler et al., 2019*) and is also effective in a neuroinflammatory model of glaucoma (*Ko et al., 2020*). While NMNAT1 gene replacement is a potential treatment option for LCA9, if SARM1 plays a more general role in retinal degeneration, then using gene therapy to express this dominant negative SARM1 could not only treat LCA9, but also multiple retinal neurodegenerative diseases. In addition, SARM1 is an enzyme and so small molecule enzyme inhibitors would be another attractive treatment modality (*DiAntonio, 2019*; *Krauss et al., 2020*). These findings demonstrate the utility of dissecting the molecular mechanism of degeneration in diseases of retinal neurodegeneration. In the case of LCA9, these studies identified a SARM1-dependent photoreceptor cell death pathway and discovered the heretofore unknown commonality between the mechanism of retinal neurodegeneration and pathological axon degeneration.

# Materials and methods

## Key resources table

| Reagent type (species) or resource | Designation | Source or reference | Identifiers | Additional information |
|---|---|---|---|---|
| Genetic reagent (mouse, male and female) | Nmnat1 $^{FRTgeo;loxP}$ | EUCOMM | RRID:MGI:5782147 | C57BL/6J to generate NMNAT1 knockout mouse |
| Genetic reagent (mouse, male and female) | CAG-CreERT2 | The Jackson Laboratory | 004682, RRID:IMSR_JAX:004682 | Whole body Cre transgenic mouse:C57BL/6J |
| Genetic reagent (mouse, male and female) | Rho-Cre | *Li et al., 2005* | Rhodopsin- iCre75 | Rod-specific Cre transgenic mouse:C57BL/6J |
| Genetic reagent (mouse, male and female) | OPN1LW-Cre | *Le et al., 2004* | human red/green pigment-Cre, RRID:IMSR_JAX:032911 | Cone-specific Cre transgenic mouse:C57BL/6J |

*Continued on next page*

*Continued*

| Reagent type (species) or resource | Designation | Source or reference | Identifiers | Additional information |
|---|---|---|---|---|
| Recombinant DNA reagent | NMNAT1 | *Sasaki et al., 2015* | | |
| Antibody | Anti-Rhodopsin (mouse monoclonal) | Abcam | Cat# ab3267, RRID:AB_303655 | IF(1:500) |
| Antibody | anti-HA (Rabbit monoclonal) | Cell Signaling Technology | Cat# 3724, RRID:AB_1549585 | IF(1:400) |
| Recombinant DNA reagent | GAPDH mouse | This paper | PCR primers | TGCCCCCATGTTTGTGATG |
| Recombinant DNA reagent | GAPDH mouse | This paper | PCR primers | TGTGGTCATGAGCCCTTCC |
| Recombinant DNA reagent | NMNAT1 mouse | This paper | PCR primers | AGAACTCACACTGGGTGGAAG |
| Recombinant DNA reagent | NMNAT1 mouse | This paper | PCR primers | CAGGCTTTTCCAGTGCAGGTG |
| Recombinant DNA reagent | AAV8(Y733F) | *Zolotukhin et al., 2002* | | |
| Cell line (*Homo-sapiens*) | HEK293 | ATCC | CRL-1573 RRID:CVCL_0045 | Used for AAV virus production |

## Mouse

Animal studies were carried out under approved protocols from animal studies committee at Washington University. NMNAT1 mutant mice (*Nmnat1 $^{FRTgeo;loxP/+}$*) which have FRT sites flanking promoterless *LacZ-neomycin phosphotransferase* gene (*beta Geo*) expression cassette located between exon 2 and 3 together with loxP sites flanking exon three was obtained from EUCOMM (NMNAT $^{tm1a(EUCOMM)Wtsi}$, RRID:MGI:5782147). This mouse expresses functionally null truncated NMNAT1 (exon 1 and 2) fused to beta Geo. *Nmnat1 $^{FRTgeo;loxP/+}$* heterozygote mice were viable and fertile however, in consistent with former results, no whole body knockout (Nmnat1$^{FRTgeo;loxP/FRTgeo;loxP}$) was born (*Conforti et al., 2011*). Next *Nmnat1 $^{FRTgeo;loxP/+}$* mice were crossed with FLP recombinase expressing mice in the C57BL/6 J background to remove beta Geo cassette flanked by FRT sites and RD8 mutation that might affect the ocular phenotypes (*Mattapallil et al., 2012*). The resultant mice (*Nmnat1 $^{fl/+}$*) have two loxP sites flanking the third exon. Then *Nmnat1 $^{fl/+}$* mice were crossed with mice expressing inducible Cre recombinase under actin promoter (*CAG-CreERT2*, RRID:IMSR_JAX:004682) and *Nmnat1$^{fl/fl}$: CAG-CreERT2* mice were generated. All genotypes were confirmed by genomic PCR. NMNAT1 whole body knockout mice were generated by injecting 100 μg/g 4-hydroxytamoxifen (Sigma) into 6 to 8 weeks old *Nmnat1$^{fl/fl}$: CAG-CreERT2* with IP for total 10 days with 2 days rest after first 5 days injection. The last day of injection was counted as day 0 after tamoxifen injection. Mice expressing Cre recombinase (*CAG-CreERT2*) were obtained from The Jackson Laboratory. To generate mice lacking *Nmnat1* specifically from rod photoreceptors, we crossed *Nmnat1$^{fl/fl}$* mice with mice carrying a copy of the Rhodopsin- iCre75 transgene, in which Cre recombinase expression is driven by the rhodopsin promoter starting postnatally at P7, which were provided by Dr. Ching-Kang Jason Chen (*Li et al., 2005*). To generate mice lacking NMNAT1 specifically from cone photoreceptors, we crossed *Nmnat1$^{fl/fl}$* mice with mice carrying one copy of the Cre recombinase under human red/green pigment promoter (*OPN1LW*-Cre, RRID:IMSR_JAX:032911), which were provided by Dr. Yun Le (*Le et al., 2004*). SARM1 knockout mice were obtained from Dr. Marco Colonna (*Szretter et al., 2009*). NMNAT3 knockout mice were derived from ES cells (Nmnat3$^{tm1(KOMP)Mbp}$, Knockout Mouse Project (KOMP)) in our facility and crossed with C57BL/6 J mice for at least five generations.

## AAV preparation

Plasmids containing the photoreceptor-specific human rhodopsin kinase (hGRK1) promoter upstream of either GFP or HA-tagged NMNAT1 were packaged in AAV8(Y733F) capsid. The detailed methodology of vector production and purification has been previously described

(*Zolotukhin et al., 2002*). Briefly, vectors were packaged using a plasmid based system in HEK293 cells (ATCC CRL-1573, RRID:CVCL_0045) by $CaPO_4$ transfection. Cells were harvested and lysed by successive freeze thaw cycles. Virus within the lysate was purified by discontinuous iodixanol step gradients followed by further purification via column chromatography on a 5 ml HiTrap Q sepharose column using a Pharmacia AKTA FPLC system (Amersham Biosciences, Piscataway, NJ, USA). Vectors were then concentrated and buffer exchanged into Alcon BSS (sodium-155.7 mM, potassium-10.1 mM, calcium- 3.3 mM, m- 1.5 mM, chloride- 128.9 mM, citrate- 5.8 mM, acetate- 28.6 mM, osmolality- 298 mOsm) supplemented with Tween 20 (0.014%). Virus was titered by qPCR relative to a standard and stored at −80C° as previously described (*Jacobson et al., 2006*). HEK293 cells used for producing AAV were purchased directly from ATCC who applies appropriate quality controls for maintaining and confirming identification of these lines. HEK293 cells were used more than two years ago for viral production and thus could not be authenticated now. HEK293 cells are passaged 50 times before discarding culture and thawing new vial. Short tandem repeat profiling is performed annually to authenticate the cell lines.

## Subretinal injections

Mice were anesthetized with a mixture of ketamine (70–80 mg/kg) and xylazine (15 mg/kg) injected intraperitoneally. The pupil was dilated with 1% tropicamide and topical anesthesia (0.5% proparacaine hydrochloride ophthalmic solution) was also applied to the eye. A self-sealing scleral incision was made by using the tip of a 31 G needle with the bevel pointed down. Then a 33G needle on a Hamilton syringe was inserted into the scleral incision and 1 μl of AAV containing solutions were injected in the subretinal space inducing a transient retinal detachment. The needle was slowly removed to prevent reflux and an ophthalmic ointment of neomycin/polymyxin B sulfate/bacitracin zinc was applied to the injected eye.

## Fundus microscopy and fluorescent angiography

Digital color fundus photography was performed using the Micron III retinal imaging system (Phoenix Research Laboratories). Prior to fundus imaging, mice were anesthetized with an intraperitoneal injection of 86.9 mg/kg ketamine and 10 mg/kg xylazine and administered 1.0% tropicamide eye drops (Bausch and Lomb) to dilate the pupils.

## Electroretinography (ERG)

ERG was performed as previously described (*Hennig et al., 2013*) by using the UTAS-E3000 Visual Electrodiagnostic System running EM for Windows (LKC Technologies). Mice were anesthetized by intra peritoneal injection of a mixture of 86.9 mg/kg ketamine and 13.4 mg/kg xylazine. The recording electrode was a platinum loop placed in a drop of methylcellulose on the surface of the cornea; a reference electrode was placed sub-dermally at the vertex of the skull and a ground electrode under the skin of the back or tail. Stimuli were brief white flashes delivered via a Ganzfeld integrating sphere, and signals were recorded with bandpass settings of 0.3 Hz to 500 Hz. After a 10 min stabilization period, a 9-step scotopic intensity series was recorded that included rod-specific/scotopic bright flash responses. After a 5 min light adaptation period on a steady white background, a 7-step cone-specific/photopic intensity series was recorded. Scotopic and photopic b-wave amplitudes and scotopic a-wave amplitudes were recorded for all flash intensities. We extracted quantitative measurements from the ERG waveforms using an existing Microsoft Excel macro that defines the a-wave amplitude as the difference between the average pre-trial baseline and the most negative point of the average trace and defines the b-wave amplitude as the difference between this most negative point to the highest positive point, without subtracting oscillatory potentials.

## Quantitative RT-PCR

Mice were euthanized and eyeballs were enucleated and retinas were dissected and immediately freeze in liquid $N_2$. On the day of preparation, Trizol was directly added to the frozen retina and tissues were homogenized with Polytron and RNA was extracted using Trizol (Thermo Fisher Scientific) and chloroform (Sigma) phase separation. Quantitative RT-PCR reaction was performed with primers (*Nmnat1*-forward: AGAACTCACACTGGGTGGAAG, *Nmnat1*-reverse: CAGGCTTTTCCAGTGCAGG TG, Gapdh-forward: TGCCCCCATGTTTGTGATG, Gapdh-reverse: TGTGGTCATGAGCCCTTCC)

with reaction mixture (ThermoFisher, SYBR Green PCR Master Mix) and monitored with Prism 7900HT (ABI) and analyzed with delta-CT method.

## Histology

Mice were euthanized and eyeballs were enucleated and fixed in 4% formalin for 8 hr then washed with PBS and then embedded in paraffin. The thickness of the retinal layers or outer nuclear layers was measured using HE stained sections and plotted against the distance from the optic nerve head. The numbers of nuclei in the outer nuclear layer were analyzed using HE stained retinal sections. Outer nuclear layer was visually determined and the number of nucleus in each layer was counted and normalized by the length parallel to each layer of the retina. Data were expressed relative to the total number of nuclei in the WT. For immunostaining of the HA epitope tag, paraffin embedded eye sections were deparaffinized and treated with formic acid (70% in water) for 15 min at room temperature. Sections were rinsed and treated with blocking solution (goat IgG). Primary antibody against HA (Cell Signaling Technology, 3724, 1:400, RRID:AB_1549585) and secondary antibody Jackson Immuno Research Laboratories, AlexaFluo@568, 111-585-003 were used to visualize the HA-tagged NMNAT1. Primary antibody against rhodopsin (Abcam, ab3267, 1:500, RRID:AB_303655) was used to identify the photoreceptor outer segment. Slides were analyzed under the microscope (Nikon, Eclipse 80i) after the nuclear staining with DAPI and mounting (Vector Laboratories, VECTASHIELD with DAPI, H-1200–10). For X-Gal staining, retina were dissected and fixed in the cold fixation buffer (0.2% Glutaraldehyde, 5 mM EGTA, 2 mM MgCl2, 0.1M K-phoshate buffer pH7.2) for 1 hr, wash with detergent rinse (0.02% Igepal, 0.01% Sodium Deoxycholate, and 2 mM MgCl$_2$in 0.1M phosphate buffer pH 7.3), and incubated with X-Gal solution (1 mg/ml X-Gal, 0.02% Igepal, 0.01% Sodium Deoxycholate, 5 mM Potassium Ferricyanide, 5 mM Pottassium Ferrocyanide, and 2 mM MgCl$_2$,0.1M phosphate buffer pH 7.3) for 10 hr in the dark at room temperature. Tissues were rinsed with PBS then fix with 4% PFA for 1 hr then paraffin sections were prepared and the sections were analyzed under the microscope (Nikon, Eclipse 80i).

## Metabolite measurement

Mice were euthanized and eyeballs were enucleated and retinas were dissected and immediately freeze in liquid N$_2$. On the day of extraction, retinal tissues were homogenized in 160 µl of cold 50% MeOH solution in water using homogenizer (Branson) and then centrifuged (15,000 g, 4℃, 10 min). Clear supernatant was transferred to new tube containing 100 µl chloroform and vigorously shake then centrifuged (15,000 g, 4℃, 10 min). The chloroform extraction was repeated three times. Clear aqueous phase (120 µl) was transferred to new tube and then lyophilized and stored at −80℃ until measurement. Lyophilized samples were reconstituted with 60 µl of 5 mM ammonium formate (Sigma) and centrifuged at 12,000 x g for 10 min. Cleared supernatant was transferred to the sample tray. Serial dilutions of standards for each metabolite in 5 mM ammonium formate were used for calibration. Liquid chromatography was performed by HPLC (1290; Agilent) with Atlantis T3 (LC 2.1 × 150 mm, 3 µm; Waters) (*Hikosaka et al., 2014*). For steady-state metabolite analysis, 20 µl of samples were injected at a flow rate of 0.15 ml/min with 5 mM ammonium formate for mobile phase A and 100% methanol for mobile phase B. Metabolites were eluted with gradients of 0–10 min, 0–70% B; 10–15 min, 70% B; 16–20 min, 0% B (*Hikosaka et al., 2014*). The metabolites were detected with a triple quadrupole mass spectrometer (6460, Agilent) under positive ESI multiple reaction monitoring (MRM) using m/z for NAD$^+$:664 > 428, NMN:335 > 123, cADPR: 542 > 428, and Nam:123 > 80. Metabolites were quantified by MassHunter quantitative analysis tool (Agilent) with standard curves and normalized by the protein amount in the sample.

## Statistical analysis

Sample number (n) was defined as a number of mice or replicates and indicated in the figure legend. Data comparisons were performed using Mann-Whitney U test, Kruskal-Wallis test, one-way ANOVA, or two-way ANOVA using R. F and P values for ANOVA were reported for each comparison in corresponding figure legends. For multiple comparisons, Holm-Bonferroni multiple comparison for one-way-ANOVA and Tukey post-hoc test for two-way-ANOVA were used.

## Acknowledgements

We thank Amy Strickland, Nina Panchenko, Kimberly Kruse, Andrea Santeford, Rachel McClarney, Simburger Kelli, Timothy Fahrner, Neiner Alicia, and Cassidy Menendez for technical assistance.

## Additional information

### Competing interests

Yo Sasaki: YS is a consultant to Disarm Therapeutics. Aaron DiAntonio: AD is a co-founder and shareholder in Disarm Therapeutics. Jeffrey Milbrandt: JM is a co-founder and shareholder in Disarm Therapeutics. The other authors declare that no competing interests exist.

### Funding

| Funder | Grant reference number | Author |
|---|---|---|
| National Institute on Aging | AG013730 | Jeffrey Milbrandt |
| National Institute of Neurological Disorders and Stroke | NS087632 | Aaron DiAntonio Jeffrey Milbrandt |
| National Cancer Institute | CA219866 | Aaron DiAntonio Jeffrey Milbrandt |
| National Eye Institute | EY019287-06 | Rajendra S Apte |
| The Needleman Center for Neurometabolism and Axonal therapeutics | | Aaron DiAntonio Jeffrey Milbrandt |
| Edward N. & Della L. Thome Memorial Foundation | | Rajendra S Apte |
| Carl and Mildred Almen Reeves Foundation | | Rajendra S Apte |
| Starr Foundation | | Rajendra S Apte |
| Bill and Emily Kuzma Family Gift for Retinal Research | | Rajendra S Apte |
| Jeffrey Fort Innovation Fund | | Rajendra S Apte |
| Glenn Foundation for Medical Research | | Rajendra S Apte |
| Research to Prevent Blindness, Inc | Unrestricted Grant to the Department of Ophthalmology | Rajendra S Apte |
| Washington University in St. Louis Medical Scientist Training Program (NIH). | T32 GM07200 | Joseph B Lin |
| Research to Prevent Blindness Nelson Trust Award | | Rajendra S Apte |

The funders had no role in study design, data collection and interpretation, or the decision to submit the work for publication.

### Author contributions

Yo Sasaki, Conceptualization, Data curation, Formal analysis, Validation, Investigation, Visualization, Methodology, Writing - original draft, Writing - review and editing; Hiroki Kakita, Shunsuke Kubota, Abdoulaye Sene, Tae Jun Lee, Norimitsu Ban, Zhenyu Dong, Data curation, Formal analysis, Validation, Investigation, Methodology, Writing - review and editing; Joseph B Lin, Validation, Investigation, Methodology, Writing - review and editing; Sanford L Boye, Resources; Aaron DiAntonio, Supervision, Funding acquisition, Writing - original draft, Writing - review and editing; Shannon E Boye, Conceptualization, Resources, Investigation, Methodology, Writing - original draft, Writing - review and editing; Rajendra S Apte, Conceptualization, Supervision, Funding acquisition,

Investigation, Methodology, Writing - original draft, Project administration, Writing - review and editing; Jeffrey Milbrandt, Conceptualization, Supervision, Funding acquisition, Methodology, Writing - original draft, Project administration, Writing - review and editing

### Author ORCIDs
Yo Sasaki ![ORCID] https://orcid.org/0000-0003-0024-0031
Tae Jun Lee ![ORCID] http://orcid.org/0000-0003-2699-2573
Joseph B Lin ![ORCID] https://orcid.org/0000-0001-6667-9018
Sanford L Boye ![ORCID] http://orcid.org/0000-0002-8803-9369
Aaron DiAntonio ![ORCID] https://orcid.org/0000-0002-7262-0968
Shannon E Boye ![ORCID] https://orcid.org/0000-0002-7312-3197
Rajendra S Apte ![ORCID] https://orcid.org/0000-0003-2281-2336

### Ethics
Animal experimentation: This study was performed in strict accordance with the recommendations in the Guide for the Care and Use of Laboratory Animals of the National Institutes of Health. Animal studies were carried out under approved protocols from animal studies committee at Washington University protocols (# 20-0020).

### Decision letter and Author response
Decision letter https://doi.org/10.7554/eLife.62027.sa1
Author response https://doi.org/10.7554/eLife.62027.sa2

## Additional files
### Supplementary files
• Transparent reporting form

### Data availability
All data generated or analyzed during this study are included in the manuscript. Source data files have been provided for Figures 1, 2, 3, 4, Fig1 supplement1, Fig2 supplement1, and Fig3 supplement1.

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
