## [Decision Letter]

**Acceptance summary:**

[Editors’ note: the authors submitted for reconsideration following the decision after peer review. What follows is the decision letter after the first round of review.]

Thank you for submitting your work entitled "SARM1 depletion rescues NMNAT1 dependent photoreceptor cell death and retinal degeneration" for consideration by *eLife*. Your article has been reviewed by a Senior Editor, a Reviewing Editor, and three reviewers. The following individuals involved in review of your submission have agreed to reveal their identity: Rui Chen (Reviewer #3).

Our decision has been reached after consultation between the reviewers. Based on these discussions and the individual reviews below, we regret to inform you that your work will not be considered further for publication in *eLife* at this time.

Your paper on the underpinnings of Leber congenital amaurosis type 9 (LCA9), an early onset retinal neurodegenerative disease due to loss of function mutations in the nuclear NAD+ synthesis enzyme NMNAT1, was reviewed by three reviewers and me. Your study, involving a careful series of loss- and gain-of-function studies on NMNAT1 using conditional or tissue-specific NMNAT1 knockout mice, presents a new angle on mechanism of this disease in that you connect photoreceptor degeneration with the long known link between loss of NMNAT and activation of SARM1, the "executioner" of axon degeneration.

As you can see in the attached reviews, there are a number of concerns raised by the reviewers. Most notably, reviewers #1 and #2 are concerned about low sample sizes for your statistical analyses. In the consultation session with the reviewers, reviewer #3 agrees that in many cases, the number of mice is low, but finds this acceptable given the large number of different conditional knock out mice reported in this paper and that actCre, iCre, HGRPCre, and Vmd2Cre show consistent results. However, the *eLife* guidelines for proper rigor stand for addressing the sample size in your work.

Another concern is on the value of the therapeutic experiments on partial rescue by AAV-NMNAT1: You show proof-of-principle that AAV supplementation with NMNAT1 may help, but you would need to address timing in testing with regard to disease onset in order to make claims about its efficacy as a therapeutic. The therapeutic approach could be downplayed and placed with a more cogent explication of the pros and cons in the Discussion section.

Finally, reviewer #1 cites a recent paper on a Sarm1 KO promoting rod and cone photoreceptor cell survival in a model of retinal degeneration (Ozaki et al., 2020). This paper does offset the novelty of your study, but your study, if amended, has the potential to elucidate additional mechanistic aspects.

While it may be difficult to perform more experiments to increase your n in this time of lab closing and ramp-up, at the very least you may be able to address other major issues on your quantitative analysis and minor points (on histology, effect on vasculature and other retinal cells, for example) for which you may already have data.

Reviewer #1:

The study by Sasaki et al., examine the role of NMNAT1 and SARM1 in retinal degeneration.

In my view there are two aspects to the study, the first is more pre-clinical and the second explore the basis for retinal degeneration in Nmnat1 mutations. I personally find the paper somehow disappointing on both.

Following extensive characterization of the Nmnat1 cKO, the authors turn to rescue experiments by AAV infection to "test the therapeutic potential of AAV-mediated gene

replacement" as they claim. There are two problems with these experiments:

1) The authors use the conditional KO and ablate Nmnat1 two months after birth, but this a genetic disease, the complete KO is much more appropriate.

2) The AAV is injected before the ablation of Nmnat1 (Results section), this does not mimic any therapeutic approach.

The rescue is not impressive, I would like to see the wild type in the graphs of Figure 6C-E.

There is not much new on basis for retinal degeneration here as well, which pool of Nmnat1 is the protecting, nucleus or cilia is highly interesting but this is not studied. What is the type of cell death? The authors point that it is apoptotic (Results section), but Sarm induce non-apoptotic cell death.

Data rigorously

In some cases, the N is too low see Figure 1C and Figure 7F-H, this leads to conflicting results is there NAD+ depletion in the Nmnat1 cKO? In Figure 7F there is in Figure 1C maybe with more samples. The authors claim that SARM1 is not protecting through elevation of NAD+ levels, but according to Figure 7F this is exactly what is happing in the Double KO.

Current literature

Recent paper demonstrated that sarm1 KO promotes rod and cone photoreceptor cell survival in a model of retinal degeneration (Ozaki et al., 2020). It is up to *eLife* to decide if this is affecting the novelty of this study.

Reviewer #2:

Mutations in the NAD+ synthesis enzyme NMNAT1 are associated with Leber congenital amaurosis type 9 (LCA9). However, little is known about the role of NMNAT1 in the retina and in regards to retinal degeneration. The authors examine retina morphology and physiology in mice after knockout: ubiquitous, in the rods, cones or RPE cells. They find that retinal degeneration occurs strongly with the loss of NMNAT1 in the rod cells, and that AAV supplementation with NMNAT1, or SARM1 knockout, protects against the retinal degenerative phenotype. This work has high significance and the authors utilize multiple model systems to support their hypothesis. The possible targeting of SARM1 as a therapeutic is highly intriguing. However, there is some clarification needed in the manuscript and low sample sizes are concerning to fully support the suggested results.

1) The authors make a point of discussing the vascular phenotype after tamoxifen induced knockout of NMNAT1 by a ubiquitous cre-driver. However, there is no mention of this vascular phenotype in any of their conditional model systems, and it is not a phenotype typically associated with LCA. The authors should mention this within their discussion and touch upon the potential reasons for lack of this phenotype when loss of NMNAT1 occurs in the rods, cones, or RPE cells alone as opposed to ubiquitously lost within the retina.

2) As the rod and cone conditional loss of NMNAT1 displayed different phenotypic outcomes, it would be interesting to see staining of the rod and cone photoreceptors in each model system to see if they were preferentially affected. The authors provide H and E images within the manuscript and should have sections from their mutant mouse lines available.

3) The authors mention that NMNAT2 interacts with SARM1 and that NMNAT1 may be compensating for this action in the photoreceptor cell. Can the authors examine NMNAT2 levels (via western blot or immunostaining) after tamoxifen-induced loss of NMNAT1? It would be interesting to see if NMNAT2 levels change in the retina, and if NMNAT1 levels are altered in other retinal cells when lost in their conditional knockout models.

4) It would be interesting to see if NMNAT1 levels are altered in other retinal cells when lost in their conditional knockout models. Can the authors examine NMNAT1 levels (via western blot or immunostaining) in these mice?

5) LCA is an early-onset retinal degenerative disease. The conditional knockout models that were used in this study did not express cre until after one week of age in the mouse. The authors should include a small discussion on this point in their paper.

6) AAV restoration of NMNAT1 was provided before disease onset, and not delivered after tamoxifen injection. It still supports the author's hypothesis that NMNAT1 plays a key role in retinal degeneration, but the authors over-state the therapeutic effect in their paper. It would also be interesting to test the effects of the AAV on the retinal cells and not just the photoreceptors, the authors could include that in their discussion. They should make sure that they do not over-state the therapeutic benefit of the AAV injections.

7) The sample sizes provided in the figure legends do not designate whether they refer to the number of mice or number of retinas/eyes examined. Please correct.

8) The photopic ERG results in the RPE conditional knockout are lower than wild-type controls without any error bars rising above WT levels. This is one of the cases where the authors note an N = 2 for statistical analysis. The authors should consider whether an increased sample size or a later timepoint tested might provide a significant ERG phenotype in this mouse line, and that their conclusion that the RPE does not play a role may be incorrect.

9) There appear to be error bars missing from some of the ERG figures. Please check and make sure all error bars are present in the graphical results.

10) The authors should mention sample sizes for the histological analyses (H and E staining) in the figure legend or Results section. It would also be helpful to discuss whether the degeneration phenotype was variable between mice or followed a similar degeneration pattern in all mice in each of their conditional knockout models.

Reviewer #3:

In this study, the authors have generated multiple knock out mice lines to systematically investigate the function of Nmnat1 in the retina. Compared to previously published studies of Nmnat1 knock out mice, this study combines cell type-specific knock out and rescue experiment to illustrate that Nmnat1 play important function primarily in photoreceptor cells, not RPEs. In addition, as the Sram1 mutant rescues the Nmnat1 knock out phenotype along with increasing NMN level and cADPR expression in the Nmnat1 knock out, the author provides convincing evidence that degeneration induced in Nmnat1 mutation is mediated through activation of Sram1. The overall data quality is high and the result is convincing. My specific comments and suggestions are:

1) As the thickness of ONL and INL change across the retina, the reduction can be better measured by measuring the thickness at multiple positions across the retina and representing the results in a spider graph.

2) Relative minor degeneration of INL neurons is observed. I am wondering if all cell types in the INL are affected. Staining of the cell types in INL can be conducted to better characterize the degeneration.

3) It seems that there is a theme that NMNAT1 is required only for retinal neurons. If this is the hypothesis, I am wondering why RGCs are not affected in the mutant.

4) The authors should fix the gene nomenclature convention. For mouse, the gene symbols are italicized, with only the first letter in upper-case.

[Editors’ note: further revisions were suggested prior to acceptance, as described below.]

Thank you for submitting your article "SARM1 depletion rescues NMNAT1 dependent photoreceptor cell death and retinal degeneration" for consideration by *eLife*. Your article has been reviewed by Huda Zoghbi as the Senior Editor, a Reviewing Editor, and three reviewers. The reviewers have opted to remain anonymous.

The reviewers have discussed the reviews with one another and the Reviewing Editor has drafted this decision to help you prepare a revised submission.

Summary:

Your study examines mutations in the NAD+ synthesis enzyme NMNAT1 that are associated with Leber congenital amaurosis type 9 (LCA9). You follow the role of NMNAT1 in the retina with regards to retinal degeneration, determining retinal morphology and physiology in mice after ubiquitous, rod- and cone-specific knockout. You show that retinal degeneration occurs with the loss of NMNAT1 in rod cells, and that a loss of SARM1, usually studied in the context of axon degeneration, protects against the retinal degenerative phenotype. This work points to an important pathway underlying retinal degeneration and possible targeting of SARM1 as a therapeutic. The reviewers have appreciated your revisions, in particular, the increase in n, and the deletion of the pre-clinical portion of the manuscript. Your study provides insight into the basis for retinal degeneration in mutation of Nmnat1. There are a few remaining amendments requested by the reviewers.

Essential revisions:

You provide AAV restoration of NMNAT1 before disease onset in your mouse model. Although you acknowledge that you did not test the AAV supplementation after disease onset at a clinically relevant time point, you indicate in both the Results section and Discussion section that AAV gene therapy had a significant rescue of the ERG, implying that the b-wave was also rescued. However, the scotopic b-wave was not significantly different between the treated and control eyes, nor was the photopic ERG, beyond the ERG a-wave response. Please correct these statements.

The level of NMN, NAD+ and cADPR is measured in the whole retina. Given that severe degeneration of photoreceptor cell is observed at the time of measurement, it is difficult to tell how much of the change is due to cell degeneration and cell type composition change. To this end, while it is clear that survival of photoreceptor cell depends on NMNAT1, you might check if subtle defects are observed in other cell types in the retina since NMNAT1 is ubiquitously expressed in all retinal cell types, by staining with cell type specific markers.

[Editors’ note: further revisions were suggested prior to acceptance, as described below.]

Thank you for resubmitting your work entitled "SARM1 depletion rescues NMNAT1 dependent photoreceptor cell death and retinal degeneration" for further consideration by *eLife*. Your revised article has been evaluated by Huda Zoghbi (Senior Editor) and a Reviewing Editor.

The study is excellent and the manuscript has been improved, but there is one remaining issue that needs to be addressed before acceptance, as outlined below:

Please clearly state in the Materials and methods section, that the HEK293 ATCC CRL-1573 cell lines were used two years ago for viral production and thus could not be authenticated now.

---

## [Author Response]

[Editors’ note: the authors resubmitted a revised version of the paper for consideration. What follows is the authors’ response to the first round of review.]

Reviewer #1:There are two problems with these experiments:1) The authors use the conditional KO and ablate Nmnat1 two months after birth, but this a genetic disease, the complete KO is much more appropriate.

The NMNAT1 whole body knockout mouse is embryonic lethal and so we could not perform the suggested experiment. Instead, we performed a conditional deletion in adult mice and found that NMNAT1 is required for the maintenance of photoreceptors and also performed both rod and cone specific knockouts that occur much earlier, at about one week postnatally. Despite the very different pattern and timing of knockouts, we got quite consistent results from these experiments, suggesting that our findings reflect a fundamental role for NMNAT1 in photoreceptors.

2) The AAV is injected before the ablation of Nmnat1 (Results section), this does not mimic any therapeutic approach.The rescue is not impressive, I would like to see the wild type in the graphs of Figure 6C-E.

We agree that this approach does not mimic gene therapy, and in response to this comment and the editor’s comments we have dramatically toned down any discussion of this as a model of therapy. We have moved these rescue data from the main figures to the supplemental data and, instead of discussing this as a candidate therapy, now emphasize that this is a genetic rescue experiment useful for demonstrating that NMNAT1 is required in photoreceptor cells. We now explicitly describe the limitations of our experiment as a model of gene therapy.

There is not much new on basis for retinal degeneration here as well, which pool of Nmnat1 is the protecting, nucleus or cilia is highly interesting but this is not studied. What is the type of cell death? The authors point that it is apoptotic (Results section), but Sarm induce non-apoptotic cell death.

We respectfully disagree with the reviewer. Our finding that loss of NMNAT1 induces retinal degeneration via activation of SARM1 is novel and quite unexpected. In the retina field the belief based on published literature was that NMNAT1 was required for photoreceptor survival due to its role in NAD biosynthesis to supply NAD+ to photoreceptors.

Here, we show that its essential function in photoreceptor survival is to inhibit SARM1. In the axon degeneration field, it is well known that NMNAT2 inhibits SARM1 activity in the axon, but no one had previously suggested that NMNAT1 would play an analogous role in a neuronal cell body. Our findings are surprising, not intuitive, and important for both our understanding of retinal neurodegeneration and axon degeneration.

Reviewer #2:[…] However, there is some clarification needed in the manuscript and low sample sizes are concerning to fully support the suggested results.1) The authors make a point of discussing the vascular phenotype after tamoxifen induced knockout of NMNAT1 by a ubiquitous cre-driver. However, there is no mention of this vascular phenotype in any of their conditional model systems, and it is not a phenotype typically associated with LCA. The authors should mention this within their discussion and touch upon the potential reasons for lack of this phenotype when loss of NMNAT1 occurs in the rods, cones, or RPE cells alone as opposed to ubiquitously lost within the retina.

We agree that these data were ancillary to the point of the manuscript and so we removed the angiogram data.

2) As the rod and cone conditional loss of NMNAT1 displayed different phenotypic outcomes, it would be interesting to see staining of the rod and cone photoreceptors in each model system to see if they were preferentially affected. The authors provide H and E images within the manuscript and should have sections from their mutant mouse lines available.

We believe that rods and cones are preferentially affected in the corresponding KO mice for a number of reasons. First, both the Rod and HGRP Cre driver mice have been previously characterized and show specificity. Second, both our anatomical and ERG data support the selective nature of the KOs. For example, the contributions of rods and cones to the various components of the ERG are well known in the literature, and our results are consistent with selective disruption of rods and cones in the respective KO lines.

3) The authors mention that NMNAT2 interacts with SARM1 and that NMNAT1 may be compensating for this action in the photoreceptor cell. Can the authors examine NMNAT2 levels (via western blot or immunostaining) after tamoxifen-induced loss of NMNAT1? It would be interesting to see if NMNAT2 levels change in the retina, and if NMNAT1 levels are altered in other retinal cells when lost in their conditional knockout models.4) It would be interesting to see if NMNAT1 levels are altered in other retinal cells when lost in their conditional knockout models. Can the authors examine NMNAT1 levels (via western blot or immunostaining) in these mice?

Despite extensive efforts on our part, we have not found reliable antibodies for NMNAT1 or NMNAT2 for immunocytochemical analysis. We would point out, however, that even if there were a compensatory increase in NMNAT2 levels in photoreceptors after NMNAT1 depletion, this is not sufficient to block SARM1 activation.

5) LCA is an early-onset retinal degenerative disease. The conditional knockout models that were used in this study did not express cre until after one week of age in the mouse. The authors should include a small discussion on this point in their paper.

We added this discussion point as requested.

6) AAV restoration of NMNAT1 was provided before disease onset, and not delivered after tamoxifen injection. It still supports the author's hypothesis that NMNAT1 plays a key role in retinal degeneration, but the authors over-state the therapeutic effect in their paper. It would also be interesting to test the effects of the AAV on the retinal cells and not just the photoreceptors, the authors could include that in their discussion. They should make sure that they do not over-state the therapeutic benefit of the AAV injections.

We appreciate the comment and agree. As such, we moved these data to the supplement and have dramatically toned down any claims about therapy, focusing instead on the study as a proof of genetic rescue.

7) The sample sizes provided in the figure legends do not designate whether they refer to the number of mice or number of retinas/eyes examined. Please correct.

The sample number is the number of mice and this is now stated clearly in the methods and figure legends. We also now provide all source data that shows color coded data for individual mice.

8) The photopic ERG results in the RPE conditional knockout are lower than wild-type controls without any error bars rising above WT levels. This is one of the cases where the authors note an N = 2 for statistical analysis. The authors should consider whether an increased sample size or a later timepoint tested might provide a significant ERG phenotype in this mouse line, and that their conclusion that the RPE does not play a role may be incorrect.

Because we would not be able to generate more mice in a timely manner, we have removed these data from the paper. As this was a negative result and we have positive results with two other tissue specific Cre lines, we feel that this has no significant impact on the paper.

9) There appear to be error bars missing from some of the ERG figures. Please check and make sure all error bars are present in the graphical results.

All of the error bars were included, but for some data the error bars were smaller than the size of symbols. We have now increased the size of the error bars to ensure that they are visible for all figures.

10) The authors should mention sample sizes for the histological analyses (H and E staining) in the figure legend or Results section. It would also be helpful to discuss whether the degeneration phenotype was variable between mice or followed a similar degeneration pattern in all mice in each of their conditional knockout models.

As shown in the Figure 3B and its legends, we analyzed the conditional knockout retinal structure using 3 mice at each of 7 different time points (total 21 mice/genotype) and obtained the retinal degeneration time course. All mice analyzed showed a consistent retinal degeneration time course.

Reviewer #3:1) As the thickness of ONL and INL change across the retina, the reduction can be better measured by measuring the thickness at multiple positions across the retina and representing the results in a spider graph.

As requested, we now provide a spider graph showing retinal and ONL thickness and agree that this is an excellent way to represent the data. We also provide representative whole eye images that clearly show the consistent loss of ONL and INL nucleus across the entire retina.

2) Relative minor degeneration of INL neurons is observed. I am wondering if all cell types in the INL are affected. Staining of the cell types in INL can be conducted to better characterize the degeneration.

While this is an interesting question, we think that this is beyond the scope of this paper which is focused on photoreceptor degeneration and function.

3) It seems that there is a theme that NMNAT1 is required only for retinal neurons. If this is the hypothesis, I am wondering why RGCs are not affected in the mutant.

While we do not know the answer, we speculate that NMNAT2 is likely a major NMNAT in RGCs and so the loss of NMNAT1 is not sufficient to activate SARM1. Unfortunately, there are no available NMNAT2 conditional knockout mice or antibodies to test retinal expression of NMNAT2. This will be an interesting future project.

4) The authors should fix the gene nomenclature convention. For mouse, the gene symbols are italicized, with only the first letter in upper-case.

We fixed the nomenclature as requested.

[Editors’ note: what follows is the authors’ response to the second round of review.]

Essential revisions:You provide AAV restoration of NMNAT1 before disease onset in your mouse model. Although you acknowledge that you did not test the AAV supplementation after disease onset at a clinically relevant time point, you indicate in both the Results section and Discussion section that AAV gene therapy had a significant rescue of the ERG, implying that the b-wave was also rescued. However, the scotopic b-wave was not significantly different between the treated and control eyes, nor was the photopic ERG, beyond the ERG a-wave response. Please correct these statements.

We agree that our statements are not sufficient to point out that ERG recovery is partial. We corrected the statements as below.

1) Results section

Original statement

“we observed significantly increased scotopic a-wave amplitudes in AAV-NMNAT1 treated retinas compared with retinas injected with AAV-GFP (Figure 3—figure supplement 1D) accompanied by a small increase in the scotopic and photopic b-wave amplitudes between AAV-NMNAT1 and AAV-GFP treated retinas (Figure 3—figure supplement 1E, F).”

Revised statement

“we observed significantly increased scotopic a-wave amplitudes in AAV-NMNAT1 treated retinas compared with retinas injected with AAV-GFP (Figure 3—figure supplement 1D). There were also small, but statistically insignificant, increases in the scotopic and photopic b-wave amplitudes between AAV-NMNAT1 and AAV-GFP treated retinas (Figure 3—figure supplement 1E, F).”

2) Discussion section

Original statement

“Delivery of AAV8(Y733F)-hGRK1-NMNAT1 significantly improved the retinal phenotype of mice deficient for NMNAT1, however the virus was delivered prior to deletion of NMNAT1.”

Revised statement

“Delivery of AAV8(Y733F)-hGRK1-NMNAT1 prior to deletion of NMNAT1 resulted in partial improvement of the retinal phenotype, in particular the scotopic a-wave, of mice deficient for NMNAT1.”

The level of NMN, NAD+ and cADPR is measured in the whole retina. Given that severe degeneration of photoreceptor cell is observed at the time of measurement, it is difficult to tell how much of the change is due to cell degeneration and cell type composition change. To this end, while it is clear that survival of photoreceptor cell depends on NMNAT1, you might check if subtle defects are observed in other cell types in the retina since NMNAT1 is ubiquitously expressed in all retinal cell types, by staining with cell type specific markers.

We agree that the whole retinal metabolite changes could be due not only to photoreceptor degeneration but also to changes in cells other than photoreceptors. As the reviewer suggests, immunocytochemistry to shed light on the state of other cell types in the degenerating retina will provide valuable information. Because our main conclusions will not be affected by results of these experiments, we choose to provide these data as a Research Advance in *eLife* or the equivalent in the future as suggested by the editors.

[Editors’ note: what follows is the authors’ response to the second round of review.]

The study is excellent and the manuscript has been improved, but there is one remaining issue that needs to be addressed before acceptance, as outlined below:Please clearly state in the Materials and methods section, that the HEK293 ATCC CRL-1573 cell lines were used two years ago for viral production and thus could not be authenticated now.

Thank very much for the suggestion on the cell line authentication. We added the description “HEK293 cells were used more than two years ago for viral production and thus could not be authenticated now.” in the method section. We appreciate careful considerations on our manuscript.